# Synergistic Effects of the Jackfruit Seed Sourced Resistant Starch and *Bifidobacterium pseudolongum* subsp. *globosum* on Suppression of Hyperlipidemia in Mice

**DOI:** 10.3390/foods10061431

**Published:** 2021-06-21

**Authors:** Zeng Zhang, Yuanyuan Wang, Yanjun Zhang, Kaining Chen, Haibo Chang, Chenchen Ma, Shuaiming Jiang, Dongxue Huo, Wenjun Liu, Rajesh Jha, Jiachao Zhang

**Affiliations:** 1Key Laboratory of Food Nutrition and Functional Food of Hainan Province, College of Food Science and Engineering, Hainan University, Haikou 570228, China; zzeng66@163.com (Z.Z.); wang2019yuanyuan@163.com (Y.W.); c15364690101@163.com (H.C.); mcc19970104@163.com (C.M.); jsm15501859060@163.com (S.J.); 17784625873@126.com (D.H.); 2Spice and Beverages Research Institute, Chinese Academy of Tropical Agricultural Science, Wanning 571533, China; zhangyanjun0305@163.com; 3Hainan Provincial People’s Hospital, Haikou 570311, China; kainch@sina.com; 4Key Laboratory of Dairy Biotechnology and Engineering, Ministry of Education P.R.C., Key Laboratory of Dairy Products Processing, Ministry of Agriculture and Rural Affairs China, Inner Mongolia Agricultural University, Hohhot 010018, China; wjliu168@163.com; 5Department of Human Nutrition, Food and Animal Sciences, College of Tropical Agriculture and Human Resources, University of Hawaii at Manoa, Honolulu, HI 96822, USA; rjha@hawaii.edu

**Keywords:** hyperlipidemia, resistant starch, *Bifidobacterium pseudolongum*, gut microbes, synbiotics

## Abstract

Approximately 17 million people suffer from cardiovascular diseases caused by hyperlipidemia, making it a serious global health concern. Among others, resistant starch (RS) has been widely used as a prebiotic in managing hyperlipidemia conditions. However, some studies have reported limited effects of RS on body weight and blood lipid profile of the host, suggesting further investigation on the synergistic effects of RS in combination with probiotics as gut microbes plays a role in lipid metabolism. This study evaluated the effects of jackfruit seed sourced resistant starch (JSRS) as a novel RS on mice gut microbes and hyperlipidemia by performing 16s rRNA and shotgun metagenomic sequencing. The results showed that 10% JSRS had a limited preventive effect on bodyweight and serum lipid levels. However, the JSRS promoted the growth of *Bifidobacterium pseudolongum*, which indicated the ability of *B. pseudolongum* for JSRS utilization. In the validation experiment, *B. pseudolongum* interacted with JSRS to significantly reduce bodyweight and serum lipid levels and had a therapeutic effect on hepatic steatosis in mice. Collectively, this study revealed the improvements of hyperlipidemia in mice by the synergistic effects of JSRS and *B. pseudolongum*, which will help in the development of “synbiotics” for the treatment of hyperlipidemia in the future.

## 1. Introduction

Hyperlipidemia is a chronic systemic metabolic disease with lower levels of high-density lipoprotein cholesterol (HDL-C) and higher levels of total cholesterol (TC), triglycerides (TG), and low-density lipoprotein cholesterol (LDL-C) due to abnormal fat transport or metabolism [1]. It is considered as one of the risk factors of cardiovascular diseases, including atherosclerosis [2], coronary heart disease [3], and diabetes [4]. Intestinal microbes, the “second genome” of the human body [5], are inextricably linked to these diseases, and the intestinal microbiota of such patients is significantly different from that of healthy people [6]. The gut microbiome in hyperlipidemic subjects is also characterized by low microbial diversity, such as a high abundance of some taxa from the phylum Actinobacteria and lower abundance of many taxa from phyla Proteobacteria and Bacteroidetes [7]. Li et.al [8] found that *Alistipes*, *Intestinibacter*, *Subdoligranulum,* and *unidentified Ruminococcaceae* in the gut were significantly negatively correlated to TG, which was an important indicator of hyperlipidemia.

Resistant starch (RS) is defined as “the sum of starch and starch degradation products not absorbed by the small intestine of a healthy individual” [9], which is naturally found in cereal grains, seeds, heated starch, or starch-containing food [10]. RS has been widely recognized for its beneficial effects, such as improving insulin resistance and glucose homeostasis [11,12], maintaining colon health [13], controlling body weight [14], elevating large-bowel short-chain fatty acids (SCFAs) [15], and especially lowering blood lipid [16,17]. It has been suggested that a high dosage of RS administration in hamsters could increase HDL-C concentration and decrease TG, TC, and LDL-C concentrations to ameliorate hyperlipidemia [18], and *Bifidobacteria* and *Lactobacillus* in the gut were dramatically increased and positively correlated with blood lipid levels. Some of these functions of RS are linked to their fermentation characteristics, thus labeled as prebiotic. However, there are also inconsistent reports. Zhang et al. [19] tested the effect of RS on 19 volunteers and found no significant difference in body weight and HDL-C. However, LDL-C decreased significantly after four weeks of treatment. This was similar to a study reporting that body weight and liver triglycerides of C57BL/6J mice did not change after being fed a 45%-fat diet with 20% high-amylose-maize RS [20]. This might be because different types of RS are affected differently due to their fermentation characteristics, thereby affecting the gut physiology and health differently [21], which is affected by the gut microbial ecology [22]. Thus, these inconsistencies might be due to neglecting the interaction between RS and the key bacteria in the intestine. Therefore, we wanted to explore the crucial role of microbiota during the digestion of RS. Generally, jackfruit seed is a good source of RS [23]. However, there is limited or no information about the effect of jackfruit seed sourced resistant starch (JSRS) on the intestinal microbiota and hyperlipidemia, requiring exploration.

To address the problem, a two-stage experiment was conducted. In Stage Ι, we revealed the effect of JSRS on hyperlipidemia-related indexes and gut microbiome in mice. Meanwhile, *Bifidobacterium pseudolongum* was identified as the key microorganism that can utilize RS in the gut through 16s rRNA and shotgun metagenomic data. Based on this result, in vitro and in vivo validation experiments in Stage II were conducted to determine the ability of *B. pseudolongum (Bifidobacterium pseudolongum* subsp. *globosum)* to utilize JSRS and to explore the synergistic effects between the two for the prevention and treatment of hyperlipidemia in mice. This study highlights the irreplaceable role of intestinal microbes in the utilization of RS and a new idea that intervention studies of functional macromolecules, including RS, need to consider the involvement of exogenous microorganisms.

## 2. Methods

### 2.1. Animal Feeding and Diet Formula

Four-week-old C57BL/6 J male mice (*n* = 40) were sourced from Hunan SJA Laboratory Animal Co. Ltd. China, which were bred and housed in specific pathogen-free conditions with 12-h day and night light cycles at 22 ± 2 °C temperature and 55 ± 10% relative humidity. The normal-fat diet (NFD) was made up of 41% corn, 26% bran, 29% bean cake, 1% salt, 1% bone meal, 1% lysine, and 1% other. The high-fat diet (HFD) was made up of 1% cholesterol, 10% egg yolk, 10% lard, 0.2% sodium cholate, and 78.8% NFD. The proximate nutrient content of the NFD was as follows: crude protein, 18%; crude fat, 4%; crude fiber, 5%; calcium, 1.0–1.8%; and phosphorus, 0.6–1.2%. The proximate nutrient content of the HFD was as follows: crude protein, 17.6%; crude fat, 19.7%; crude fiber, 4%; calcium, 0.81–1.45%; and phosphorus, 0.56–1.04%. JSRS was provided by the Spice and Beverage Research Institute, Chinese Academy of Tropical Agricultural Sciences, Haikou, China. The *B. pseudolongum* was provided by Inner Mongolia Agricultural University, China.

### 2.2. Stage Ι, the Exploration Experiment

After a 7-day adaptation period, mice were randomly divided into 4 groups: NFD (*n* = 10); NFD plus JSRS (90% NFD plus 10% JSRS, *n* = 10); HFD (*n* = 10); HFD plus JSRS (90% HFD plus 10% JSRS, *n* = 10). Body weight was assessed at weeks 1, 2, 4, and 8. Feces were collected at weeks 2, 4, 8, and stored at −80 °C until further microbiota analysis. After 8 weeks of intervention, all mice were subjected to a 16-h fast and then euthanized and dissected. Blood was collected, and serum was isolated to determine TG as a blood lipid indicator. All fecal samples from week 2 and week 4 were used for high-throughput sequencing of the V3-V4 region of the bacterial 16s rRNA gene [24], and fecal samples from week 8 were processed for deep metagenomic sequencing [25]. The experimental design is presented graphically in Figure 1A.

### 2.3. Stage II, the Validation Experiment

In vitro validation. *B. pseudolongum* was inoculated into MRS agar medium (agar 20 g, glucose 20 g, peptone 10 g, beef extract 10 g, yeast extract 5 g, C_6_H_5_O_7_(NH_4_)_3_ 2 g, Tween-801 mL, CH_3_COONa 5 g, K_2_HPO_4_ 2 g, MgSO_4_ 0.58 g, MnSO_4_ 0.25 g, water 1 L) as the control and JSRS agar medium (agar 20 g, JSRS 20 g, peptone 10 g, beef extract 10 g, yeast extract 5 g, C_6_H_5_O_7_(NH_4_)_3_ 2 g, Tween-80 1 mL, CH_3_COONa 5 g, K_2_HPO_4_ 2 g, MgSO_4_ 0.58 g, MnSO_4_ 0.25 g, water 1 L) anaerobic culture for 48 h, and the total number of microbial colonies were calculated. To determine the extent of starch utilization by *B. pseudolongum*, the plate was treated with iodine plus potassium iodide solution, and diameter of the colony and transparent circle was calculated.

In vivo validation. After 2 weeks of acclimatization on an NFD to the laboratory environment, mice (*n* = 40) were randomly subdivided equally into 4 groups [NFD, (*n* = 10); TR (HFD was fed for the first 3 weeks for making the obese mice model, then, 90% NFD plus 10% JSRS plus 8Log CFU *B. pseudolongum* infusions were done later, *n* = 10); HFD (*n* = 10); PR (90% HFD plus 10% JSRS plus 8Log CFU *B. pseudolongum* infusions, *n* = 10)] and were treated for 3 weeks to make the nutritionally obese mice model. After 3 weeks, three mice from each group were euthanized to determine abdominal fat accumulation. The rest of the mice were continued on the respective dietary treatment for another 3 weeks. The NFD and HFD groups were gavaged with an equal volume of normal saline as controls (Figure 1B). Body weight was recorded weekly. Feces were collected weekly and stored at −80 °C. After 8 weeks of intervention, mice were subjected to a 16-h fast and then were euthanized. Livers and abdominal fat were excised and weighed quickly. Then, livers were washed by phosphate buffer saline and processed for further analysis.

### 2.4. Scanning Electron Microscopy

The JSRS particle structure (granule form) analysis was done using scanning electron microscopy (SEM) as described by Zhang et al. [26] with some modifications. The samples were fixed on a sample holder with a silver plate and coated with a platinum film. The obtained samples were observed under a scanning electron microscope (Verios G4 UC, Thermo Fisher Scientific, Brno, South Moravia, Czech Republic).

### 2.5. Serum Lipid Levels Determination

The blood from the eyeball was collected and centrifuged at 571× *g* for 15 min at 4 °C. Commercial assay kits (Jian Cheng Biotechnology Co., Ltd., Nanjing, China) were used to measure blood lipid indices, including TG, TC, HDL-C, and LDL-C. The results of hepatic lipids were corrected for total protein concentration.

### 2.6. Liver Histology and Morphometric Assessment

The largest lobe of the fresh liver from each mouse in the different treatment groups was fixed in paraformaldehyde at room temperature and embedded in paraffin. Five-micrometer-thick sections of the liver tissue were cut, dewaxed, stained with hematoxylin-eosin (H&E), dehydrated, and sealed with neutral gum. To determine possible histopathological changes, the slides were then observed under an upright optical microscope (Nikon, Shanghai, China) for routine morphological evaluation and image acquisition (Servicebio Technology Co., Ltd., Wuhan, China).

### 2.7. DNA Extraction and High throughput Metagenomic Sequencing

Total DNA was extracted from fecal samples by using the QIAamp^®^ DNA Stool Mini Kit (Qiagen, Hilden, Germany) according to the manufacturer’s instruction, and purity and integrity of DNA were determined by 0.8% agarose gel electrophoresis. The concentration of DNA was determined by NanoDrop 2000 (Novogene Company, Beijing, China). The V3-V4 region of the 16s ribosomal RNA (rRNA) gene was amplified as previously described [24]. The shotgun metagenomic sequencing was performed by Illumina HiSeq 2500 instrument (Novogene, Beijing, China). About 150 bp paired-end gene fragments were obtained by sequencing, and a sequencing library with the length of the DNA fragments of 300 bp was prepared.

### 2.8. Bioinformatics for Amplification and Shotgun Metagenomic Sequencing Analysis

According to the quality scores, the sequences were filtered by using the sliding window approach. The low-quality sequences for which the average quality score over a 50-nt sliding window dropped below 30 were removed from the raw sequences. The QIIME (v1.9) platform was used for bioinformatics analysis [27]. The Operational Taxonomic Units (OTU) were selected by QIIME (default settings). OTUs were classified with a 97% threshold identity. An RDP Classifier was used to obtain the taxonomy against the RDP database using the representative [28]. The metagenomic reads were trimmed using Sickle software and subsequently aligned to the mouse genome (GRCm38.p6) to remove the host DNA fragments for subsequent analysis. For metagenomic species annotation, MetaPhlan2 software for taxonomic classification was employed [29]. HUMAnN2 was performed for metagenomic functional features and metabolic-pathway annotation based on the UniRef90 database [29].

### 2.9. Statistical Analysis and Figure Construction

The statistical analyses were performed using R. Data are presented as mean ± standard error (SE). The differential abundances of the genera were tested using the Wilcoxon rank-sum test and the Kruskal–Wallis test, and *p* < 0.05 were considered to be significantly different genera. Principal coordinate analysis (PCoA) was performed using the “ade4” package. The package “ggplot2” was used for generating line charts, box plots, and bar charts. The heatmap was constructed using the “pheatmap” package. The *p*-value was filtered and corrected by the “DESeq2” package in the bubble diagram. The microbial ecological networks were inferred by Spearman’s rank correlation coefficient from the metagenomic sequencing data and visualized in Cytoscape (Version 3.7.1).

## 3. Results

### 3.1. The Granules Form of JSRS

The morphology of JSRS granules is shown in Appendix A. The SEM images of JSRS samples showed that the starch granules were mainly semi-oval or bell-shaped. The particle size of starches was about 10 μm, which was similar to our previous finding [26].

### 3.2. 10% JSRS Could neither Significantly Restrain the Body Weight Gain nor Maintain Serum Lipid Levels

There was no significant difference in body weight among treatment groups (*p* > 0.05) at the beginning of the study (0 week). After eight weeks, the body weight of the HFD group and HFD plus JSRS group were significantly higher than that of the NFD group. There was no significant difference between the HFD and HFD plus JSRS groups (Appendix A). Moreover, the TG levels were significantly lower in the NFD and NFD plus JSRS groups as compared with the HFD group, while the HFD plus JSRS group was significantly higher than that of the NFD group (Appendix A).

### 3.3. The JSRS Regulated Dysbiosis in Intestinal Microbiota Caused by a High-Fat Diet

The alpha and beta diversity of the mice gut microbes in different groups were compared among the second, fourth, and eighth weeks based on the high-throughput sequencing data of 16s rRNA. In the fourth week’s results, the points representing NFD and HFD groups had different clustering trends in the PCoA plots (Figure 2A), which indicated that the HFD could significantly affect the gut microbes. In the eighth week, the alpha diversity index of the JSRS group remained at a high level, and the Shannon index of the NFD plus JSRS group was significantly higher than the NFD group (Appendix A). The gut microbe structure of the HFD plus JSRS group was more similar to that of the NFD group, compared with that of the HFD group at week eight (Figure 2B), which implied the addition of JSRS to the HFD made it possible to correct the HFD-induced intestinal dysbiosis.

### 3.4. Bifidobacterium Pseudolongum Was the Potential Functional Intestinal Microorganism in JSRS Utilization

Since there was a difference in the intestinal microbiota between NFD and HFD groups, the differential microorganisms at the genus levels in JSRS utilization were found on Mann–Whitney tests by combined NFD plus JSRS group and HFD plus JSRS group into the JSRS group. The remaining two groups were combined into the NFD and HFD group (N and H group), based on the high-throughput 16s rRNA sequencing abundance results at week 2 and week 4. Differential microorganisms at the species level in the eighth week were calculated by the same method, as the abundance results were derived from metagenomic sequencing data.

In the fourth week, the genus abundance of *Bifidobacterium*, *Parabacteroides*, *Akkermansia*, *Robinsoniella,* and *Unclassified Beijerinckiaceae* in the JSRS group was significantly higher than that in the N and H group, while the *Bifidobacterium* and *Parabacteroides* were also significantly different in the second week. The abundance of *B. pseudolongum*, *Akkermansia muciniphila*, and *Subdoligranulum unclassified* were significantly higher in the JSRS group than in the N and H group at week eight (Figure 3A). Overall, the average abundance of *Bifidobacterium* was significantly higher than that of the N and H group at weeks two and four, which highlighted the importance of the species belonging to the *Bifidobacterium* genus. Accordingly, for fecal samples collected at week eight, shotgun metagenomic sequencing was applied to identify the potential specific species of *Bifidobacterium*, which utilized JSRS in mice guts. Metagenomic species annotation results confirmed that *B. pseudolongum* was the dominant species capable of utilizing JSRS for their growth in mice gut (Figure 3A). Using the same grouping method, the top twenty metabolic pathways with significant baseline differences were screened (Figure 3B). Compared with the JSRS group, 6-hydroxymethyl-dihydropterin diphosphate biosynthesis Ι, anaerobic energy metabolism, anhydro-muropeptides recycling, chorismate biosynthesis from 3-dehydroquinate, L-lysine biosynthesis, Purine nucleobases degradation, superpathway of N-acetylglucosamine, N-acetylmannosamine, N-acetylneuraminate degradation, and Superpathway of phospholipid biosynthesis were significantly more enriched in the N and H group. The remaining 12 pathways such as L-tryptophan biosynthesis and superpathway of pyrimidine ribonucleotides de novo biosynthesis were enriched in JSRS group. Collectively, the major differential metabolic pathways were more active in the JSRS group. JSRS significantly altered specific metabolic pathways in mice gut microbes.

### 3.5. The Ability of B. pseudolongum to Utilize the JSRS In Vitro

Since we found *B. pseudolongum* in mice gut to be strongly correlated with JSRS intake, to test this hypothesis, we performed an in vitro validation experiment in Stage II. A strain of *B. pseudolongum* from the gut of cattle was used in an in vitro validation experiment. The glucose in the MRS agar medium was replaced entirely by JSRS (JSRS agar medium), and *B. pseudolongum* was inoculated. After 48 h, the average growth of *B. pseudolongum* in the starch agar medium reached 6Log CFU/mL, although the average growth of the control group was 7Log CFU/mL. The ratio of JSRS utilization circle reached 8.3 (Appendix A). These results proved that *B. pseudolongum* could grow by using JSRS as the primary carbon source in vitro.

### 3.6. The Synergistic Effect of JSRS and B. pseudolongum Suppressed Hyperlipidemia in Mice

Based on the results above, another hypothesis in the present study is that the synergistic effect of JSRS and *B. pseudolongum* could suppress hyperlipidemia in mice. Therefore, we performed an in vivo validation experiment to test this hypothesis (Figure 1B). At week three, there was no significant difference in body weight between the TR and HFD groups, both of which were significantly higher than the NFD group. The body weight of the TR group was significantly higher than the NFD group (Figure 4A), although there was no significant difference in body weight between the two groups. After another three weeks, the body weight of the TR group not only gradually decreased but also was lower than the NFD group (Figure 4B). Moreover, the body weight of the PR group was consistently lower than the HFD group, even lower than the NFD group. The synergistic effect of JSRS and *B. pseudolongum* prevented high fat diet-induced obesity in mice.

Data on serum lipid levels and abdominal fat of mice further verified that the synergistic effect could reduce abdominal fat and blood lipid in mice. At week six, with the synergistic effect of both, the abdominal white fat weight was effectively reduced and controlled in PR and TR groups, which was significantly lower than that in the HFD group. TR group was slightly higher than the NFD group (Figure 4C), but no significant differences existed. The same trend was found in the anatomy of mice (Appendix A). Similarly, the level of blood lipids of mice reinforced the conclusion that JSRS and *B. pseudolongum* worked together to reduce body weight and serum lipid levels in mice. The levels of TC, TG, and LDL showed the highest trend in the HFD group (Figure 4D). The levels of these three blood lipid parameters were significantly reduced by the synergistic effect, although there was no significant difference in HDL-C level between the four groups. TC and TG levels were lower in the TR group than in the NFD group, while they were not statistically significantly different in LDL-C levels between the two groups.

The histopathology results showed that the liver structure of the NFD group was normal, and the liver lobules were clearly visible. The liver cells were intact, without rupture or autolysis, and closely arranged. No fatty vacuoles were observed. In contrast, the liver of the HFD group mice had many fatty vacuoles and disturbed hepatic cord distribution. Hepatocytes were loosely arranged and had fuzzy margins. Evidently, the HFD caused severe pathological changes in the liver of mice. The fatty infiltration of hepatocytes was significantly reduced in the TR group, although the arrangement was slightly disturbed. Compared with the HFD group, the number of fatty vacuoles in the liver of the PR group was significantly reduced, and the hepatocytes had a clearer contour and tighter structure (Figure 4E).

### 3.7. The Potential Mechanism of the Synergistic Effect of JSRS and B. pseudolongum for Improving the Symptom of Hyperlipidemia in Mice

The previous analysis revealed the differential bacteria and metabolic pathways. To better reveal the correlation of JSRS, metagenomic species, microbial metabolic pathways, and serum lipid indicators, we calculated Spearman’s rank correlation coefficients based on the data from Stage I of the experiment and constructed correlation networks for them (Figure 5). *B. pseudolongum*, *Subdoligranulum unclassified,* and *Akkermansia muciniphila* were positively correlated with JSRS, and *B. pseudolongum* had the highest correlation with JSRS compared to the other two bacteria. The adenosylcobalamin salvage from cobinamide I (M) as well as TG was positively correlated with *B. pseudolongum;* conversely, the thiamin salvage (E), putrescine biosynthesis (P), and superpathway of polyamine biosynthesis (S) were negatively correlated. The network diagram showed the interconnections and interactions among JSRS, differential strains, differential metabolic pathways, and TG.

## 4. Discussion

The results of this study show that only 10% of JSRS had a limited preventive effect against hyperlipidemia in mice. On the other hand, the combination of JSRS and *B. pseudolongum* improved not only body weight, abdominal white fat, and serum lipid levels, but also had preventive and therapeutic effects on hepatic steatosis in liver cells. In previous studies, JSRS was able to regulate blood glucose levels and lipid metabolism [30], and prevent and treat obesity [31]. However, it has also been shown that the effects of JSRS on body weight and visceral fat were limited [12,32]. Similarly, this study shows that the addition of 10% JSRS to diet did not significantly reduce body weight and serum lipid levels in mice. The possible reason for this result may be that a high-fat diet could minimize the RS fermentation, resulting in lower levels of propionic acid and butyric acid [33]. SCFAs are closely related to the host’s health, especially some metabolic diseases [34,35]. This condition led to an attenuated beneficial effect of RS in rats on a 20% high-fat diet, compared to a low-fat diet [33]. Another possible reason may be the structural differences of RS. It has been reported that the RS with lower crystallinity and double helix content had a poor effect on the body weight and serum TC content of mice induced by a high-fat diet. Resistant starch with high crystallinity and double helix content had a more stable structure, resulting in a more stable and slow fermentation in the intestine, allowing the substance to be evenly distributed in the intestine and meeting the needs of the distal colon [17]. Moreover, the variation of RS polymorphism would also lead to ecological changes in the microbial community structure of the colon. Different types of RS produce slightly different proportions of SCFAs and might undergo different fermentation patterns in the gut of obese mice, resulting in different effects on intestinal health [36]. However, our results indicated that JSRS ingestion could significantly elicit a response from *B. pseudolongum*. Furthermore, we observed that the major differential metabolic pathways such as L-tryptophan biosynthesis, gluconeogenesis III and thiamin salvage II were more active in the JSRS group. It has been shown that L-tryptophan can regulate immunity [37] and intestinal homeostasis [38]. Proper nutritional supplementation with tryptophan can prevent or reduce inflammation of the gut [39]. Bacterial fermentation of fiber can improve glycemic control by producing succinate and thus activating intestinal gluconeogenesis [40]. The thiamin, as vitamin B1, is a cofactor for many enzymes indispensable for glucose and energy metabolism [41]. Apparently, JSRS can significantly improve the metabolism of intestinal microorganisms in mice and may be involved in related pathways such as host immunity and energy metabolism. Based on these findings, we hypothesized that the synergistic effect of JSRS and *B. pseudolongum* could prevent hyperlipidemia in mice. The hypothesis was then validated in the in vitro and in vivo experiments in Stage II. JSRS is not only a prebiotic but also a carbon source that promotes the growth of certain beneficial microorganisms in the gut. The pairing of prebiotics and probiotics is also gradually emerging in many studies. Ma et al. [25] found that constant supplementation with low galactose improves the stability of the intestinal microbes. Thus, probiotics may be able to work more reliably or better in the gut if they are supported by their substrates. Numerous studies have shown that the combined effects of probiotics and prebiotics can positively impact a wide range of human diseases, such as enhancing Disease Activity Score (DAS 28) and visual analog scores in rheumatoid patients, modulating plasma nitric oxide, rising glutathione [42], improving some symptoms of type 2 diabetes and quality of life [43], prolonging life, and reducing liver cancer cell proliferation in mice with malignant leukemia [44]. Research of synbiotics is expected to be promoted more with the updated definition of synbiotics [45]. *Bifidobacteria* are major degraders of RS, producing butyrate during the breakdown of RS to improve host health [46,47], and RS is the prebiotic capable of regulating changes in gut microbes [48]. There is no doubt that the combination of JSRS and *B. pseudolongum* as potential “synbiotics” can have a far more positive impact on the gut and health of mice than only JSRS supplementation.

In this research, differential microbes capable of utilizing JSRS in the intestine of mice on HFD and HFD were found. Interestingly, although 10% JSRS did not significantly improve body weight and serum TG level in mice, it was still able to regulate certain critical intestinal microbes, such as *Akkermansia*, *Ruminococcus,* and *Bifidobacterium*, which could promote the production of SCFAs that affect metabolic diseases positively to the host [49,50].

## 5. Conclusions

JSRS maintained the homeostasis of the intestinal microbes by correcting the damaging effects of a high-fat diet on gut microbes. As expected, *B. pseudolongum*, which are differential microorganisms at weeks two, four, and eight, could use JSRS to grow both in vitro and in vivo, and exerted excellent synergistic effects to treat and prevent hyperlipidemia and had a positive effect on hepatic steatosis in mice. Collectively, this study provided evidence on the suppression of hyperlipidemia in mice by the synergistic effects of JSRS and *B. pseudolongum*, which can support the development of “synbiotics” for the treatment of hyperlipidemia in the future. Collectively, this study provided evidence on the suppression of hyperlipidemia in mice by the synergistic effects of JSRS and *B. pseudolongum*, which can support the development of “synbiotics” for the treatment of hyperlipidemia in the future. By understanding the effects of JSRS on the gut microbiome and hyperlipidemia, it will help to obtain better applications in different fields, and provide a reference for the functional development of new varieties in the starch industry. We suggest the application of JSRS in combination with *B. pseudolongum* (synbiotics) to target the intestinal microbiota to alleviate hyperlipidemia in order to achieve the high value utilization of JSRS as a functional health food.

## Figures and Tables

**Figure 1 foods-10-01431-f001:**
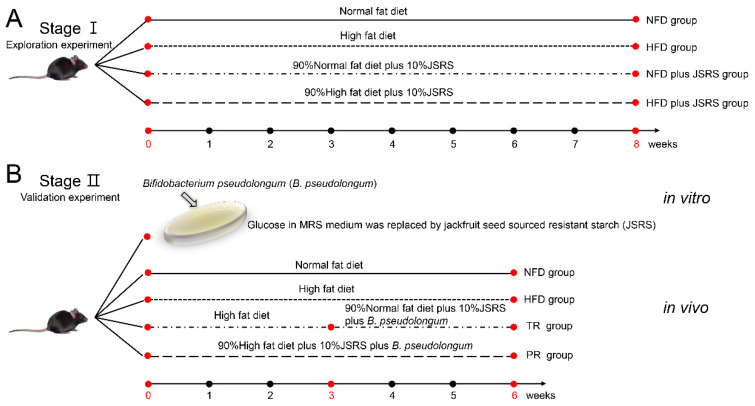
Experimental design. (**A**) In stage Ι, mice were randomly divided into four groups: NFD (normal-fat diet, *n* = 5); NFD plus JSRS (90% normal-fat diet plus 10% jackfruit seed sourced resistant starch, *n* = 6); HFD (high-fat diet, *n* = 6); HFD plus JSRS (90% high-fat diet plus 10% jackfruit seed sourced resistant starch, *n* = 6). The mice were kept for eight weeks. (**B**) The 4 groups that were treated for 3 weeks in order to make nutritionally obese mice model: NFD (normal-fat diet, *n* = 10); TR (HFD was chosen for the first 3 weeks for the development of obese mice model, then, 90% NFD plus 10% JSRSplus 8Log CFU *B. pseudolongum* infusions done later *n* = 10); HFD (high-fat diet, *n* = 10); PR (90% HFD plus 10% JSRS plus 8Log *B. pseudolongum* infusions, *n* = 10). After 3 weeks, three mice in each group were euthanized to observe abdominal fat accumulation. Then a 3-week intervention was continued on respective dietary treatments until the end of the experiment.

**Figure 2 foods-10-01431-f002:**
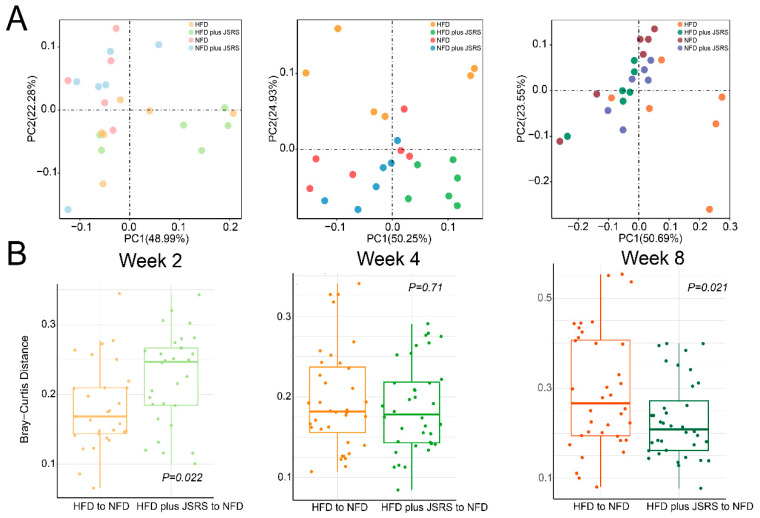
Changes in microbial structure and similarity at different time points. (**A**) Principal coordinate analysis (PCoA) showed the changes in microbial composition based on bray-curtis distance among four groups at weeks 2, 4, and 8 in Stage Ι. Each point represents the composition of the microbiota of one sample. (**B**) Comparison of the similarity between HFD group to NFD group and HFD plus JSRS group to NFD group based on Bray–Curtis distance at week 2, 4, and 8 in Stage Ι. Lower values indicate similarity to the NFD group. (JSRS, Jackfruit seed sourced resistant starch).

**Figure 3 foods-10-01431-f003:**
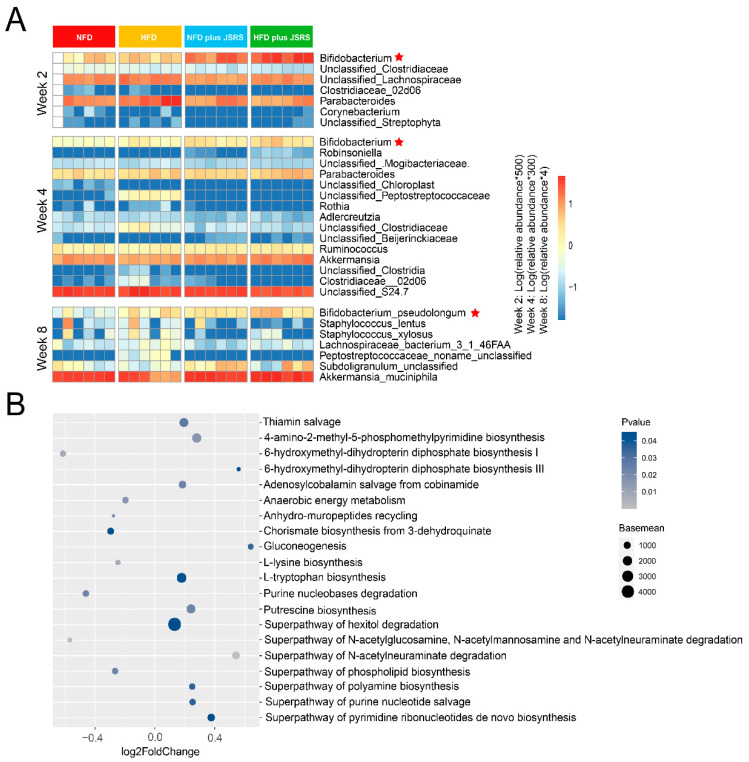
Comparison of differential microbial and metabolic pathways. (**A**) Variations of differential bacteria in different groups at weeks 2, 4, and 8 in Stage Ι. The significant difference genera (week 2 and week 4) and species (week 8) were screened (Wilcoxon rank-sum test). The depth degree of color represents the relative abundance of the genus or species (blue indicates a small number, and red indicates a large number). (**B**) Metabolic pathway analysis was performed based on the shotgun metagenomic sequencing data in Stage Ι. Significantly different pathways with baseline ranking in the top 20 were screened out. Log_2_ foldchange value greater than 0 indicated that the enrichment abundance in the JSRS group (the combination of the HFD plus JSRS group and the NFD plus JSRS group) is greater than that in the N and H group (the combination of the NFD group and the HFD group). (JSRS, Jackfruit seed sourced resistant starch).

**Figure 4 foods-10-01431-f004:**
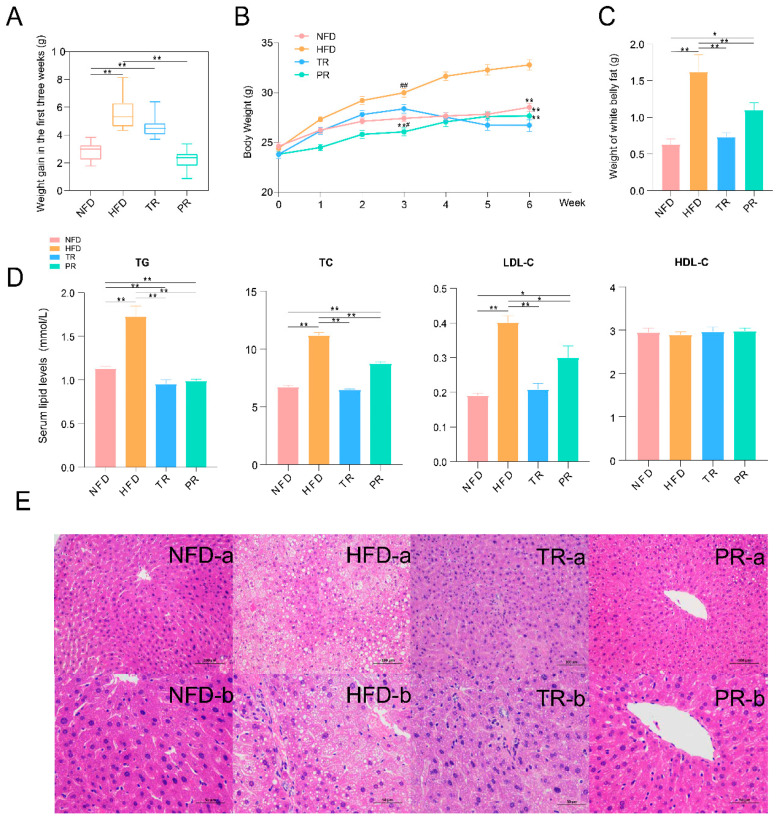
Effect of Jackfruit seed sourced resistant starch (JSRS)plus *Bifidobacterium pseudolongum* on hyperlipidemia in mice. (**A**) Body weight of four groups at each time point in Stage II. (Wilcoxon rank-sum test, * *p* < 0.05, ** *p* < 0.01 compared with HFD; ^#^ *p* < 0.05, ^##^ *p* < 0.01 compared with NFD, error bar: mean ± SE) (**B**) Weight gain was compared between the four groups during the first three weeks of the study at each time point in Stage II (Wilcoxon rank-sum test, * *p* < 0.05, ** *p* < 0.01, NS means no significant difference, error bar: mean ± SE) (**C**) Comparison of the weight of white fat in the abdomen of four groups of mice. (Wilcoxon rank-sum test, * *p* < 0.05, ** *p* < 0.01, error bar: mean ± SE) (**D**) Comparison of TC, TG, LDL-C, and HDL-C levels among the four groups in Stage II. (Wilcoxon rank-sum test, * *p* < 0.05, ** *p* < 0.01, error bar: mean ± SE) (**E**) Histopathological analysis of the liver sections of mice in four groups at 200× and 400× magnification in Stage II. (“-a” and “-b” were liver section images at the same position, but with different magnification. “-a” means 200×; “-b” means 400×).

**Figure 5 foods-10-01431-f005:**
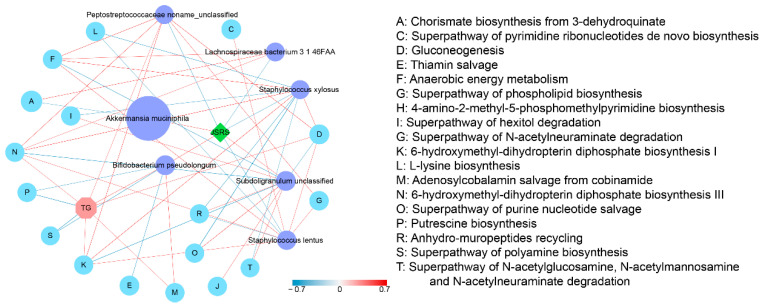
The correlation network among Jackfruit seed sourced resistant starch (JSRS), differential species, metabolic pathways, and TG. The R-value less than −0.4 or greater than 0.4 was selected. The width and color (red indicates a positive correlation, while blue indicates a negative correlation) of the edge are proportional to the correlation intensity. The green node represents JSRS, purple nodes represent species, light blue nodes represent metabolic pathways, and light red nodes represent TG. The node size is proportional to the mean abundance in the respective population.

## Data Availability

The sequence data reported in this paper have been deposited in the NCBI database (metagenomic sequencing data: PRJNA669624, PRJNA669572).

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
