# Peer review of "Synergistic Effects of the Jackfruit Seed Sourced Resistant Starch and Bifidobacterium pseudolongum subsp. globosum on Suppression of Hyperlipidemia in Mice"

_foods, 2021, doi:10.3390/foods10061431_

Round 1

Reviewer 1 Report

Zhang and colleagues reported a work entitled “Synergistic effects of the jackfruits seed sourced resistant starch and Bifidobacterium pseudolongum on suppression of hyperlipidemia in mice”. 

The authors have evaluated the effect of jackfruit (as a source of resistant starch) on the mice gut microbes and hyperlipidemia. In addition, the synergistic effect of both jackfruit and a strain belonging to B. pseudolongum on the mice was showed.
The topic of the work is of interest and in agreement with the current objectives of research regarding the symbiotic products. The research design is well designed, the methods are adequately described and the results are clearly presented. 

In my opinion the work needs some minor revision: 

  • The name of strain of B. pseudolongum used in the experiments should be reported. Consequently, the title should be arranged. 
  • I suggest improving the discussion section, especially as regards the part relating to the results obtained from the correlation between the metabolic pathway of the intestinal microorganisms of mice and jackfruits.

Minor remarks

  • Zoom in on the symbols of statistical significance in the figures.
  • Check the space in the title of paragraph 3.2
  • The authors only reported the microbial load (6 Log CFU/mL) of B. pseudolongum after 48 h of JSRS fermentation. Please report the initial inoculum concentration.
  • Par. 3.6. Please rearrange the order of sentences according to the figure panel. The sentences regarding the figure 4C should precede those regarding the figure 4D.
  • Please check “prebiotics and prebiotics” in line 369
  • Please check the figure 5 and figure 5 caption. In the figure the JSRS node is green, in the caption it is reported as yellow.

Author Response

In my opinion the work needs some minor revision: 

Point 1: The name of strain of B. pseudolongum used in the experiments should be reported. Consequently, the title should be arranged. 

Response 1: Thank you for pointing this out. In this study, Bifidobacterium pseudolongum subsp. globosum was used. We have revised it in the relevant places.

Point 2: I suggest improving the discussion section, especially as regards the part relating to the results obtained from the correlation between the metabolic pathway of the intestinal microorganisms of mice and jackfruits.

Response 2: Thank you for pointing this out. We have added the related discussions, as follows (Lines 364-373):

Also, we observed that the major differential metabolic pathways such as L-tryptophan biosynthesis, gluconeogenesis III and thiamin salvage II were more active in the JSRS group. It has been shown that L-tryptophan can regulate immunity [38] and intestinal homeostasis [39]. Proper nutritional supplementation with tryptophan can prevent or reduce inflammation of the gut [40]. Bacterial fermentation of fiber can improve glycemic control by producing succinate and thus activating intestinal gluconeogenesis [41]. The thiamin, as vitamin B1, is a cofactor for many enzymes indispensable for glucose and energy metabolism [42]. Apparently, JSRS can significantly improve the metabolism of intestinal microorganisms in mice and may be involved in related pathways such as host immunity and energy metabolism.

Reference

  1. Gao, J.; Xu, K.; Liu, H.; Liu, G.; Bai, M.; Peng, C.; Li, T.; Yin, Y. Impact of the Gut Microbiota on Intestinal Immunity Mediated by Tryptophan Metabolism. Front Cell Infect Microbiol 2018, 8, 13. [http://doi.org/10.3389/fcimb.2018.00013]
  2. Platten, M.; Nollen, E.A.A.; Rohrig, U.F.; Fallarino, F.; Opitz, C.A. Tryptophan metabolism as a common therapeutic target in cancer, neurodegeneration and beyond. Nat Rev Drug Discov 2019, 18, 379-401. [http://doi.org/10.1038/s41573-019-0016-5]
  3. Krautkramer, K.A.; Fan, J.; Backhed, F. Gut microbial metabolites as multi-kingdom intermediates. Nat Rev Microbiol 2021, 19, 77-94. [http://doi.org/10.1038/s41579-020-0438-4]
  4. De Vadder, F.; Kovatcheva-Datchary, P.; Zitoun, C.; Duchampt, A.; Backhed, F.; Mithieux, G. Microbiota-Produced Succinate Improves Glucose Homeostasis via Intestinal Gluconeogenesis. Cell Metab 2016, 24, 151-157. [http://doi.org/10.1016/j.cmet.2016.06.013]
  5. Hayashi, M.; Nosaka, K. Characterization of Thiamin Phosphate Kinase in the Hyperthermophilic Archaeon Pyrobaculum calidifontis. J Nutr Sci Vitaminol 2015, 61, 369-374. [http://doi.org/DOI 10.3177/jnsv.61.369]

Minor remarks

Point 3: Zoom in on the symbols of statistical significance in the figures.

Response 3: Thank you for pointing this out. We have enlarged the symbols of statistical significance in the figures.

Point 4: Check the space in the title of paragraph 3.2

Response 4: We apologize for this typo. We have added spaces and checked the full text carefully. (Line 196)

Point 5: The authors only reported the microbial load (6 Log CFU/mL) of B. pseudolongum after 48 h of JSRS fermentation. Please report the initial inoculum concentration.

Response 5: Perhaps the unprecise description caused your confusion. In fact, in the in vitro experiments of this study, we counted the number of colonies by solid medium. Under normal growth conditions, the number of colonies after 48 h represents the number of colonies at the time of inoculation. In in vitro experiments, 100 ul of B. pseudolongum fluid was diluted and coated in JSRS agar medium (Glucose in MRS medium was completely replaced by JSRS). The number of colonies of B. pseudolongum and JSRS utilization circles were counted after incubation at 37°C for 48 h. Also, we made a control with normal MRS agar medium (0 JSRS). The average colony count of the control group reached 6.1×7 Log CFU/mL. The purpose of this result was to demonstrate that B. pseudolongum could grow by using JSRS as the primary carbon source in vitro. We have modified the relevant descriptions in the in vitro experiments and added the results of the control group as follows (Lines 108-111 and 269-272):

Lines 108-111:

2.3. Stage Ⅱ, the validation experiment

In vitro validation. B. pseudolongum was inoculated into MRS agar medium (agar 20 g, glucose 20 g, peptone 10 g, beef extract 10 g, yeast extract 5 g, C6H5O7(NH4)3 2 g, Tween-801 ml, CH3COONa 5 g, K2HPO4 2 g, MgSO4 0.58 g, MnSO4 0.25 g, water 1L) as the control and JSRS agar medium (agar 20 g, JSRS 20 g, peptone 10 g, beef extract 10 g, yeast extract 5 g, C6H5O7(NH4)3 2 g, Tween-80 1 ml, CH3COONa 5 g, K2HPO4 2 g, MgSO4 0.58 g, MnSO4 0.25 g, water 1L) anaerobic culture for 48 hours, and the total number of microbial colonies were calculated. To determine the extent of starch utilization by B. pseudolongum, the plate was treated with iodine plus potassium iodide solution and diameter of the colony and transparent circle was calculated.

Lines 269-272:

The glucose in the MRS agar medium was replaced entirely by JSRS (starch agar medium), and B. pseudolongum was inoculated. After 48 h, the average growth of B. pseudolongum in the starch agar medium reached 6Log CFU/ml, although the average growth of the control group was 7Log CFU/ml.

Point 6: Par. 3.6. Please rearrange the order of sentences according to the figure panel. The sentences regarding the figure 4C should precede those regarding the figure 4D.

Response 6: Thank you for pointing this out. We have modified the order and checked the full text carefully. We modified this part of the sentences as follows (Line 287-295):

Data on serum lipid levels and abdominal fat of mice further verified that the synergistic effect could reduce abdominal fat and blood lipid in mice. At week 6, with the synergistic effect of both, the abdominal white fat weight was effectively reduced and controlled in PR and TR groups, which was significantly lower than that in the HFD group. TR group was slightly higher than the NFD group (Figure 4C), but no significant differences existed. The same trend was found in the anatomy of mice (Figure S3B). Similarly, the level of blood lipids of mice reinforced the conclusion that JSRS and B. pseudolongum worked together to reduce body weight and serum lipid levels in mice. The levels of TC, TG, and LDL showed the highest trend in the HFD group (Figure 4D).

Point 7: Please check “prebiotics and prebiotics” in line 369

Response 7: We apologize for this typo. We have changed the first "prebiotics" to "probiotics" and checked the full text carefully. (Line 381)

Please check the figure 5 and figure 5 caption. In the figure the JSRS node is green, in the caption it is reported as yellow.

Response 8: We apologize for this typo. We have modified the description of the color of JSRS and checked the full text carefully. (Line 337)

Reviewer 2 Report

The article is interesting and contains a correct description of the experiment. I am only asking for the title of the drawings to be included in the supplement and to re-edit the "applications" (especially lines 388 and 389)

Author Response

Point 1: The article is interesting and contains a correct description of the experiment. I am only asking for the title of the drawings to be included in the supplement and to re-edit the "applications" (especially lines 388 and 389)

Response 1: Thank you for pointing this out. The titles of the supplementary materials have been added to the manuscript as follows (Lines 416-419):

Supplemental Material Figure S1. Effects of Jackfruit seed sourced resistant starch (JSRS) on body weight and serum lipid levels of mice in Stage Ⅰ. Supplemental Material Figure S2. Alpha diversity analysis at different time points among the four groups in Stage Ⅰ.

Point 2: re-edit the "applications" (especially lines 388 and 389)

Response 2: Thank you for pointing this out. We have re-edited the "Application" as follows (Lines 406-414):

Collectively, this study provided evidence on the suppression of hyperlipidemia in mice by the synergistic effects of JSRS and B. pseudolongum, which can support the development of "synbiotics" for the treatment of hyperlipidemia in the future. By understanding the effects of JSRS on the gut microbiome and hyperlipidemia, it will help to obtain better applications in different fields and provide a reference for the functional development of new varieties in the starch industry. We suggest the application of JSRS in combination with B. pseudolongum (synbiotics) to target the intestinal microbiota to alleviate hyperlipidemia in order to achieve the high value utilization of JSRS as a functional health food.
